# Running Head: Heat Affects Cholesterol and Bile Acid Alterations in Cholesterol and Bile Acids Metabolism in Large White Pigs during Short-Term Heat Exposure

**DOI:** 10.3390/ani10020359

**Published:** 2020-02-23

**Authors:** Wei Fang, Xiaobin Wen, Qingshi Meng, Lei Liu, Jingjing Xie, Hongfu Zhang, Nadia Everaert

**Affiliations:** 1State Key Laboratory of Animal Nutrition, Institute of Animal Sciences, Chinese Academy of Agricultural Sciences, Beijing 100193, China; fangwei1987@126.com (W.F.); 13693686549@163.com (X.W.); mengqingshi@caas.cn (Q.M.); swina2010@163.com (L.L.); 2Precision Livestock and Nutrition Unit, Gembloux Agro-Bio Tech, TERRA Teaching, and Research Unit, Liège University, Passage des Déportés 2, 5030 Gembloux, Belgium; nadia.everaert@uliege.be

**Keywords:** heat stress, cholesterol, bile acids, gene expression, growing pigs

## Abstract

Heat stress influences lipid metabolism independently of nutrient intake. It is not well understood how cholesterol and bile acid (BA) metabolism are affected by heat stress. To investigate the alterations of cholesterol and bile acids when pigs are exposed to short term heat stress, 24 Large White pigs (63.2 ± 9.5 kg body weight, BW) were distributed into one of three environmental treatments: control conditions (CON, 23 °C with ad libitum intake; *n* = 8), heat stress conditions (HS, 33 °C with ad libitum intake; *n* = 8), or pair-fed conditions (PF, 23 °C with the same amount to the feed consumed by the HS; *n* = 8) for three days. Compared with CON pigs, HS pigs reduced the average daily feed intake and average daily gain by 55% and 124%, respectively, and significantly increased rectal temperatures by 0.9 °C and respiration rates more than three-fold. The serum total cholesterol (TC), low-density lipoprotein-cholesterol, and triglycerides (TG) increased (*p* < 0.05), while hepatic TC, TG, and mRNA of 3-hydroxy-3-methylglutaryl-CoA reductase were reduced on day 3. Furthermore, liver taurine-conjugated BAs (TCBAs), including taurolithocholic acid, taurochenodeoxycholic acid (TCDCA), tauroursodeoxycholic acid, taurohyodeoxycholic acid, and taurocholic acid were elevated in HS pigs compared to CON and PF pigs (*p* < 0.05), and the level of chenodeoxycholic acid was more significant in the PF group than in the CON and HS groups. The concentration of ursodeoxycholic acid in the serum was higher in HS pigs than CON and PF pigs (*p* < 0.05), and TCDCA was increased in HS pigs compared with PF pigs (*p* < 0.05). Altogether, short-term HS reduced hepatic cholesterol levels by decreasing cholesterol synthesis, promoting cholesterol to TCBAs conversion, and cholesterol release to serum in growing pigs. This independently reduced feed intake might serve as a mechanism to protect cells from damage during the early period.

## 1. Introduction

Reduced growth performance and increased economic losses in pigs caused by heat stress (HS) have been well documented in animal agriculture [1,2]. Without heat abatement strategies, growing pigs are particularly susceptible to HS [3], which has also been shown to decrease feed intake [4], alter metabolism [5], and tissue accretion [6] compared to thermoneutral-reared pigs.

Heat stress impacts lipid metabolism in pigs, mainly by decreasing the lipolytic capacity and increasing fat deposition and triglyceride storage, independent of the heat-induced inadequate feed intake [5,7]. Altered insulin sensitivity is suggested to be responsible for enhanced lipid deposition. Reduced lipolysis activity in adipose tissue appears to be an adaptive form to limit heat production in heat-stressed animals [8,9,10]. Lipids play an essential role in regulating several biological processes involved in heat stress response [11]. Cholesterol is an important lipid component, plays a unique role among the many lipids in the mammalian. Besides, heat stress induces the fluidization of cellular membranes, which play an essential role in the control of heat sensing and signaling to maintain cellular function [12,13]. Cholesterol is an essential component of most cell membranes, regulating the fluidity of the lipid bilayer and the heat shock protein signaling and contributing to heat adaptation [14,15]. Therefore, determining the effect of HS on cholesterol metabolism is an important step in the further understanding of the role of different lipids in HS adaptation. The research from Pearce et al. showed that heat-stressed pigs tended to increase serum circulating cholesterol in growing pigs [6]. However, until now, data regarding the regulation of cholesterol metabolism in heat-stressed pigs has not been entirely available. The liver is the primary organ for the biosynthesis, absorption, and secretion of cholesterol. Cholesterol is synthesized in the liver from acetyl coenzyme A, 3-hydroxy-3-methylglutaryl-CoA reductase (HMGCR) is the rate-limiting enzyme of the pathway. Furthermore, hepatocytes also take up circulating cholesterol from the serum by a low-density lipoprotein receptor (LDLR) mediated endocytosis [16,17,18]. Additionally, the sterol response element binding protein (SREBP) transcription factors regulates cholesterol biosynthesis and uptake [19].

Bile acids are main end-products of cholesterol catabolism in the liver and their synthesis is the major route for cholesterol elimination [20]. Beyond being an essential emulsifier for the intestinal absorption of lipids and fat-soluble vitamins, bile acids also are signal molecules that activate the nuclear receptor farnesoid X receptor (FXR) and the plasma membrane-bound bile acid receptor (TGR5) to regulate glucose and lipid metabolism as well as cholesterol homeostasis. Biosynthesis of primary BAs is initiated by cholesterol 7α-hydroxylase (CYP7A1) and mitochondrial sterol 27-hydroxylase (CYP27A1), which represent classic and alternative pathways, respectively. Before being secreted into the gallbladder by bile salt export pump (BSEP), primary BAs are conjugated with glycine or taurine under the catalysis of BA coenzyme A synthetase (BACS) and BA amino acid transferase (BAAT). After enterohepatic circulation, recycled BAs are taken up into hepatocytes by organic anion transporting peptides (OATPS) and sodium taurocholate cotransporting polypeptide (NTCP) respectively [21,22]. Bile acid biosynthesis in the liver is tightly regulated by feedback signaling through FXR/small heterodimer partner (SHP) signaling pathway [20].

Short and long term HS elicit different metabolic responses [8,23]. We have previously shown that long-term heat exposure decreased cholesterol uptake and taurine-conjugated BA synthesis and uptake in the liver [24,25]. However, whether the cholesterol and BA metabolism are affected at the early stages of heat stress remains to be defined. Therefore, the objective of the present study is to investigate the effect of short-term heat exposure on hepatic cholesterol and BA metabolism and to clarify the underlying mechanism via the measurement of the expression of genes involved in cholesterol and BA metabolism.

## 2. Materials and Methods

### 2.1. Animals and Study Design

All procedures involving animal usage and care have been approved by the Institute of Animal Sciences, Chinese Academy of Agricultural Sciences (IASCAAS). This experiment used 24 Large White barrows from 8 litters (3 pigs/litter, which were individually penned and equally assigned to the three treatments; the control (CON, 23 ± 1 °C), heat-stressed (HS, 33 ± 1 °C), and pair-fed (PF, 23 ± 1 °C) groups. The average initial weight of the pigs was 63.2 ± 9.5 kg and the treatments lasted for 3 days. Over the three-day experimental period, pigs from the CON and HS groups consumed feed ad libitum, while the PF pigs were given the same amount of feed consumed by the HS pigs on the previous day. All pigs were free access to water. The pigs were distributed to 6 chambers (*n* = 4 pigs/chamber). All six chambers are identical in size, design, and utilities. All chambers were automatically controlled for temperature and relative humidity. The environmental condition was at 55% ± 5% relative humidity (RH) with a 16 h light cycle (light from 6.00 A.M. to 10.00 P.M.). Before the start of the experiment, pigs were allowed to adapt to the chambers for 1 week under CON conditions. When the experiment began, the temperature of the chambers in the HS group was raised and kept at 33 ± 1 °C from day 1 to day 3 all the time, where it was maintained at 23 ± 1 °C for the CON and PF groups throughout the experimental period.

Feed was formulated according to the nutritional requirements suggested by NRC (2012). The diet formulas (Table 1) came from a previous description [24,25]. All pigs were individually fed twice a day and the feed intake (FI) of each pig was recorded. Pigs were weighed at the beginning and end to calculate the average daily gain. At 12, 36, and 60 h of the experiment, the respiration rate (RR) was calculated by counting flank movements that equated to breath per min. Rectal temperature (RT) was taken every day by mercury thermometer. Feed was withheld from the pigs 12 h before slaughter. Blood samples were withdrawn from the jugular vein before pigs were sacrificed using electrical stunning followed by exsanguinations. Serum was then obtained, aliquoted, and stored at −80 °C for lipid analysis and BA quantification. Tissue from the liver was quickly removed, snap-frozen in the liquid nitrogen, and stored at −80 °C for BA quantification and mRNA measurements.

### 2.2. Serum Lipid Profiles

Serum was obtained by centrifugation at 4 °C for 15 min at 850× *g*, and stored in aliquots at −80 °C. Serum total cholesterol (TC), triglycerides (TG), high-density lipoprotein coupled cholesterol (HDL-C), and low-density lipoprotein-coupled cholesterol (LDL-C) were measured using commercial reagent kits, as described by Wen et al. [24].

### 2.3. Hepatic Lipid Analyses

Approximately 100 mg of liver tissue was homogenized with nine times physiological saline, and the supernatant was harvested after centrifuged at 850× *g* for 10 min at 4 °C for lipid analysis. Hepatic total protein (g/protein/L, A045-2-2, Nanjing jiancheng), TC (mmol/g protein), TG, (mmol/g protein), HDL-C (mmol/g protein), and LDL-C (mmol/g protein) were quantified in the experiment.

### 2.4. Bile Acids Extraction and Quantification

The measurements of BAs extraction and quantification from serum and liver were done using LC-MS/MS following procedures as previously described [25]. Briefly, a volume of 200 μL liver homogenate or 100 μL serum was mixed with an equal amount of sodium acetate buffer (4 °C, 50 mM, pH 5.6) and triple ethanol. The mixture was incubated at 4 °C for 30 min and centrifuged at 20,000 g for 20 min. Supernatants were recovered and diluted four times with sodium acetate buffer and passed through Bond Elute C18 cartridges (Agilent 12102161, Harbor City, CA, USA), which were previously activated with 5 mL methanol. After rinsing with 25% ethanol and then 5 mL methanol for eluting bile acids. The solvent was removed under nitrogen gas and the residue was resuspended in 1 mL of methanol and filtered with 0.45 μm nylon syringe filters before injection. Thermo Dionex Ultimate 3000 high-performance liquid chromatography coupled with Waters Xevo TQ LC/MS mass spectrometer with an ESI source was used for BAs quantification. The concentrations of individual BAs were summed to derive the primary, secondary, glycine and taurine conjugated bile acids, as well as total bile acids.

### 2.5. Isolation of RNA and Quantification of mRNA Levels in the Liver

Total RNA was isolated from the liver using the RNeasy Mini Kit (Qiagen 74104, Hilden, Germany). cDNA was generated using the High-Capacity cDNA Archive kit (Takara RR047A, Dalian, China) and was subjected to quantitative RT-PCR amplification using SYBR Green Master Mix (Takara RR420A, Dalian, China). Primers for critical genes related to cholesterol and BAs synthesis, transportation, and regulation were performed as described previously [24,25]. The comparative CT method (2^−ΔΔCt^) was referred to calculate the mRNA expression levels and using β-actin as a housekeeping gene control.

### 2.6. Statistical Analysis

All the statistical analyses were performed using the JMP10.0 (SAS Institute, Inc., Cary, NC, USA). One-way analysis of variance (ANOVA) and Duncan’s multiple comparisons were used to compare the treatments. Data were presented as mean and SEM. A *p* value < 0.05 was considered significant.

## 3. Results

### 3.1. Pig Performances after Short Term Heat Stress

Pigs exposed to 33 °C for three days lost BW (−233.33 g, *p* < 0.01, Figure 1A) while CON and PF groups gained BW 823.81 g and 125 g, respectively. Average daily gain (ADG) and average daily feed intake (ADFI) was significantly decreased by 55% (*p* < 0.01, Figure 1B) compared to the CON pigs kept at 23 °C. ADG and ADFI in PF pigs were reduced to 80.3% and 44.1% (*p* < 0.01, Figure 1A,B) of the CON group, respectively. The pigs in the HS group showed a significant increase (*p* < 0.01, Figure 1C) in RT (40.1 vs. 39.2 vs. 38.7 °C) and RR (108 vs. 32 vs. 23 bpm, *p* < 0.01, Figure 1D) compared with CON and PF pigs. RT and RR did not differ (*p* > 0.05) between CON and PF pigs.

### 3.2. Serum Lipid Levels

In Table 2, serum TC, LDL-C, and TG were higher in the HS pigs than the CON and PF pigs, but HDL-C did not differ among the three groups.

### 3.3. Changes in Liver Cholesterol Biosynthesis

Hepatic TC was reduced in HS pigs (*p* = 0.047) relative to other pigs by 16.2% and 18.3%. TG was significantly lower (12.5%, *p* = 0.02) in HS group than CON group (Figure 2A). However, HDL-C and LDL-C were not significantly affected among the three groups. The expression of HMGCR in the HS pigs was almost 50% of that in the CON and PF pigs (*p* = 0.019), however, LDLR and SREBP2 were comparable among all treatments (Figure 2B).

### 3.4. Serum BAs Composition after Short Term Heat Stress

Bile acids from serum were profiled (Figure 3). Pigs with HS increased the concentration of SBA (*p* = 0.01, Figure 3A). However, there was no difference in the PBA, TCBA, and glycine-conjugated BAs (GCBA). However, ursodeoxycholic acid (UDCA, *p* < 0.05) and TCDCA (*p* = 0.035) were increased in the HS group compared with other groups and PF group, respectively (Figure 3B).

### 3.5. Hepatic BAs Composition after Short Term Heat Stress

Liver TCBAs (*p* < 0.01, Figure 4A), including tauroursodeoxycholic acid (TUDCA, *p* < 0.01), taurohyodeoxycholic acid (THDCA, *p* < 0.01), taurohyocholic acid (THCA, *p* < 0.01) and taurolithocholic acid (TLCA, *p* = 0.017) as well as TCDCA (*p* = 0.013) were significantly increased (Figure 4B) in HS pigs compared with other pigs. Greater chenodeoxycholic acid (CDCA, *p* = 0.037) was found in the PF group than CON and HS groups (Figure 4B).

### 3.6. Bile Acids Synthesis, Transporters, and Regulation in the Liver

Given the significant effect of heat exposure on liver TCBAs levels, differential hepatic expression of genes involved in BAs synthesis (CYP7A1, CYP27A1, CYP8B1, BAAT, and BACS), transports (BSEP, NCTP, and OATP1B3), and regulation of the processes (FXR, TGR5, SHP, FGFR4, and KLβ) was determined (Figure 5). Unexpectedly, no significant changes in gene expression levels were found in the liver of the HS or PF group compared with the CON group after 3 d of heat exposure.

## 4. Discussion

The pigs exposed to 33 °C resulted in marked heat stress, as evidenced by elevated rectal temperatures and respiration rates and the decrease in feed intake, which is consistent with previous reports [26,27]. Besides, short-term heat stress altered the metabolism of cholesterol and BAs in growing pigs via reduced hepatic cholesterol synthesis and promoted cholesterol release to serum and increased the level of TCBA in the liver.

A significant increase in serum TC, LDL-C, and TG were observed in the HS pigs after 3 d of heat exposure. The results are in agreement with our previous study, in which TC, LDL-C, and TG were increased on the first three days of heat exposure and gradually returned to the baseline afterward [24]. An elevated temperature changes the lipid composition and architecture of cell membranes, leading to an increase in membrane fluidify, which could be offset by increasing cholesterol levels [28]. Indeed, cholesterol levels in the membrane rise with an increase in body temperature to stabilize membranes [29,30]. Cress and Gerner also reported that cholesterol could regulate the survival and sensitivity of mammalian cells to high temperature via changes in physical membrane properties [31], serving as a primary defense mechanism for upregulation of expression of heat shock proteins (HSPs) [14] that promotes the survival of stressed cells [32], leading to resistance to heat stress [33]. The elevation in serum cholesterol in our study might have served as a primary defense mechanism against the cellular damage and noxious effects during the early stage of heat exposure.

Different from serum cholesterol, liver TC and TG were reduced in the HS pigs, together with decreased expression of HMGCR, which is a rate-limiting enzyme for cholesterol synthesis, demonstrating liver cholesterol synthesis was down-regulated in the HS pigs on day 3. The activity of HMGCR can be regulated by cholesterol in a negative feedback mode in order to maintain a relatively stable level of cholesterol [34]. It is likely that increased serum cholesterol at the beginning of heat exposure down-regulated liver HMGCR expression to reduce liver cholesterol biosynthesis. These results suggest that short term HS reduced hepatic cholesterol synthesis and promoted cholesterol release to serum in pigs. The increase of serum TG and decrease of hepatic in HS pigs indicated fat deposition. Although the PF pigs consumed equal amounts of feed as the HS pigs, the restricted feed intake did not affect serum cholesterol and liver HMGCR on day 3. On the basis of the differences between the HS and PF pigs, it is evident that cholesterol metabolism is uniquely regulated in the pigs exposed to heat stress.

Bile acids are synthesized from cholesterol in the liver and play an essential role in cholesterol homeostasis. In the current study, heat exposure of pigs led to a substantial increase in the hepatic concentration of TLCA, TCDCA, TUDCA, THDCA and THCA, and of TCBAs, independent of reduced feed intake. TCBAs are formed through conjugation taurine with bile acids and the concentration of taurine is a major determinant of the proportion of TCBAs in the liver [35,36]. Taurine could increase the amount of TCBAs by the liver in man [37]. The greater TCBAs concentrations in the liver suggested an increase of taurine in the HS pigs since other BAs were not changed in our current study. The increase of heat dissipation is a significant mechanism for coping with short term heat stress [38]. The mammalian overall body temperature is regulated by the hypothalamus, which contains high levels of taurine. Taurine is considered an endogenous refrigerant and is involved in the central mechanism of thermoregulation [39]. Possibly, taurine was released to counteract the resulting hyperthermia.

In addition to its cooling function, taurine also can alleviate the body damage caused by HS [40,41,42]. Moreover, UDCA could also protect mitochondria against reactive oxygen species (ROS) production in stress response [43,44], and exposure to heat could enhance ROS production and induce oxidative stress. Thus, during 3 d of heat exposure, increased TCBA in the liver, and UDCA in the serum could be important to increase heat loss to prevent life-threatening damages. Unexpectedly, the expression of genes involved in BA synthesis and conjugation (CYP1A1, CYP21A1, CYP8B1, BAAT, and BACS) in the liver remained unaffected in HS pigs. Bile acid synthesis is a negative feedback regulation mechanism, and FXR plays a prominent role. In the liver, it was found that FXR induces the expression of SHP, which in turn represses CYP7A1, the rate-limiting enzyme for bile acid synthesis. In our study, the expression of FXR and SHP was also unchanged, demonstrating that TCBAs synthesis was not altered in the liver after short term HS. The liver can also uptake TCBAs by NTCP to maintain the balance of bile acids [45]. However, the mRNA of NTCP in the liver was not changed, suggesting that the uptake of TCBAs in the liver was not found in heat stress pigs. Therefore, increasing the concentration of TCBAs may occur at the functional level rather than at the transcriptional level. Besides liver synthesis and uptake, TCBAs can be reabsorbed in the intestine. Thus, increased TCBAs levels in the liver seem to suggest an increased reabsorption of TCBAs from the intestine. However, work on intestinal BAs under heat stress needs to be done to validate these hypotheses in the future.

Bile acids are end-products of cholesterol metabolism and facilitate cholesterol elimination in the liver. Taurocholate infusion produced a marked increase both in total output and in the esterification of cholesterol [46]. Furthermore, Murakami S et al. concluded that taurine supplementation increases the synthesis of TCBAs and stimulates the catabolism of cholesterol to BAs, leading to reductions in hepatic cholesterol levels [47]. Thus, increased levels of TCBAs in the liver in heat-stress pigs may promote the conversion from cholesterol to BAs and consequently increase cholesterol and triglycerides release into the systemic circulation, leading to reduced cholesterol and triglycerides levels in the liver.

An inconsistent metabolic mechanism has been exhibited in short and long term heat stress. The HS response is more intense in the first three days’ response to environmental changes and seems to decline in the long-term stage to adapt to stress [26,48]. We have also found that alterations of cholesterol and BA metabolism were different between short and long-term exposure in this and previous study. In our current study, increased concentrations of cholesterol in the serum and TCBAs in the liver under short term HS indicated a potential role to prevent cellular damage, suggesting a defense mechanism against noxious effects from early stages of heat stress. It was found that a new metabolism could be established after 8 to 10 days of heat exposure [38]. Our previous data showed that long term heat exposure (21 d) did not alter serum cholesterol levels, but reduced hepatic cholesterol and concentrations of TCBAs in the serum and liver in growing pigs by suppressing cholesterol uptake and BAs synthesis, conjugation uptake, and signaling [24,25]. Taurine impacted lipid profiles by lowering the levels of TC, LDL-C, and TG in plasma in long-term models [49,50]. Indeed, taurine supplementation resulted in the recovery to control levels of elevated serum lipids after 14 days of heat exposure [51]. Moreover, Morales et al. reported that the absorptive serum concentration of taurine was decreased when pigs were exposed to heat stress for 21 days [52]. Hence, the reduction of TCBAs in the serum and liver might decrease the internal absorption of taurine, which increased serum cholesterol homeostasis in long term HS pigs. Reduced feed intake in HS results in nutritional deficiencies that damages the intestinal epithelial cells by a decreased height of the intestinal villi of HS pigs [53] and then affects nutrient absorption [54]. Thus, decreased cholesterol levels in the liver suggest that long-term heat stress reduced cholesterol uptake from extrahepatic tissues. Over prolonged heat exposure, decreased cholesterol and TCBAs levels in the liver might contribute to altering the lipid metabolism, due to the decrease of feed intake for an extended period and the increase of the insulin concentration [55] may be closely related to HS-adaptation mechanism. The adaptation condition was achieved in the body, owing to the alteration of the metabolism induced by long-term heat exposure, which might adopt an extended period of heat stress as an adaptive and compensatory capability.

Heat stress typically reduces feed intake in swine and influences the BA levels [56]. In our study, a decrease in CDCA in the liver was observed in the PF pigs after 3 d feed restriction. This result was reversed in mice, which showed that 40% calorie restriction increased hepatic CDCA concentration by 146% [57]. Diet in pigs is different from that of mice. The natural diet of mice has phytosterols, but not cholesterol, and pigs are omnivores, so they eat cholesterol. Therefore, the different dietary composition might lead to a discrepancy in the bile acid pool between pigs and mice under restricted feeding conditions.

## 5. Conclusions

In conclusion, short-term heat exposure reduced cholesterol synthesis and promoted cholesterol release to serum and conversion to TCBAs in the liver. Increased serum cholesterol and liver TCBAs levels, independent of reduced feed intake, may serve as a mechanism to prevent cellular damage during short term heat stress.

## Figures and Tables

**Figure 1 animals-10-00359-f001:**
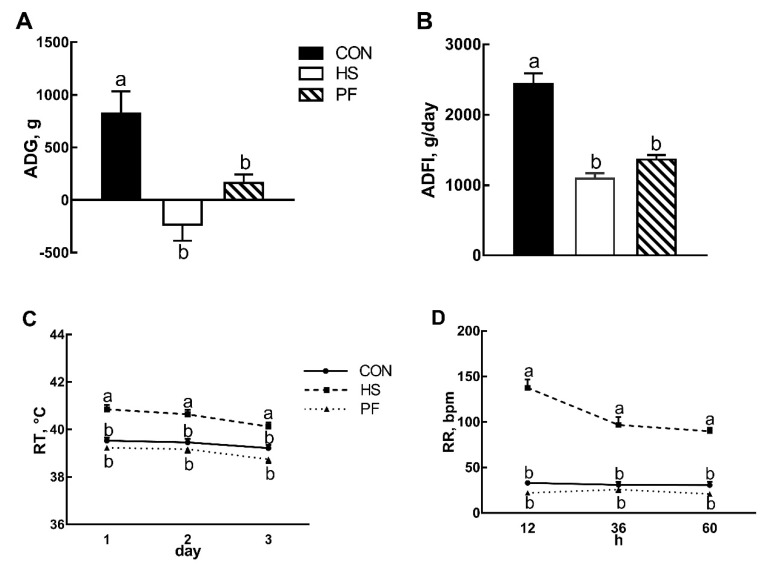
Effects of optimum condition (CON), short-time heat exposure (HS) and pair-fed group (PF) on (**A**) average daily gain, (**B**) average daily feed intake, (**C**) rectal temperature, and (**D**) respiration rates. All data are expressed as mean ± SEM and values with the same letter superscripts mean no significant difference (*p* > 0.05), while with different letter superscripts mean significant difference (*p* < 0.05). CON, 23 °C with ad libitum intake; HS, 33 °C with ad libitum intake; PF, 23 °C with the same amount to the feed consumed by the HS; ADG, average daily gain; ADFI, average daily feed intake; RT: rectal temperatures; RR: respiration rates; bpm: breaths per minute.

**Figure 2 animals-10-00359-f002:**
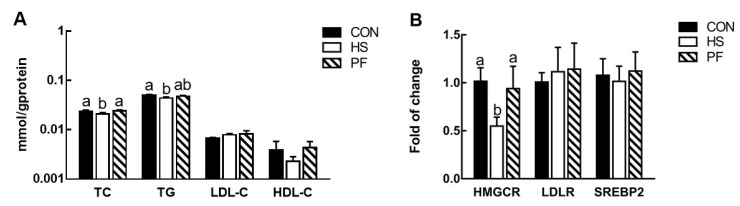
Hepatic lipids composition (**A**) and genes related to cholesterol biosynthesis, uptake and regulation (**B**) in the liver with short-term heat exposure. Values are expressed as means ± SEM of data from 8 individual tissue samples. Values with the same letter superscripts mean no significant difference (*p* > 0.05), while with different letter superscripts mean significant difference (*p* < 0.05). CON, 23 °C with ad libitum intake; HS, 33 °C with ad libitum intake; PF, 23 °C with the same amount to the feed consumed by the HS.

**Figure 3 animals-10-00359-f003:**
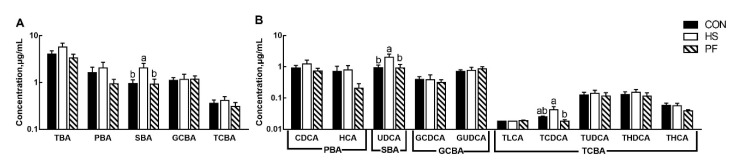
Alteration of BAs profiles in the serum after short term heat exposure. Concentrations of total, primary, secondary, glycine-conjugated and taurine-conjugated bile acids in the serum in growing pigs (**A**); concentrations of individual bile acids in the serum (**B**). All data are expressed as mean ± SEM. Values with the same letter superscripts mean no significant difference (*p* > 0.05), while values with different letter superscripts mean significant difference (*p* < 0.05). CON, 23 °C with ad libitum intake; HS, 33 °C with ad libitum intake; PF, 23 °C with the same amount to the feed consumed by the HS; BAs, bile acids; TBA, total BAs; PBA, primary BAs, SBA; secondary BAs; GCBA, glycine-conjugated BAs; TCBA, taurine-conjugated BAs; CDCA, chenodeoxycholic acid; HCA, hyocholic acid; UDCA, ursodeoxycholic acid; GCDCA, glycochenodeoxycholic acid; GUDCA, glycoursodeoxycholic acid; TLCA, taurolithocholic acid; TCDCA, taurochenodeoxycholic acid; TUDCA, tauroursodeoxycholic acid; THDCA, taurohyodeoxycholic acid; THCA, taurohyocholic acid.

**Figure 4 animals-10-00359-f004:**
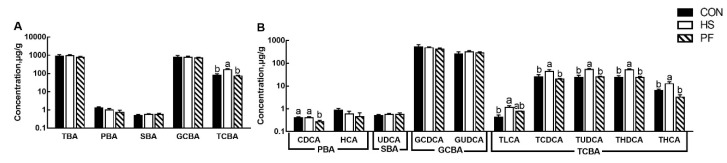
Alteration of BAs profiles in the liver after short term heat exposure. Concentrations of total, primary, secondary, glycine-conjugated, and taurine-conjugated bile acids in the liver in growing pigs (**A**); concentrations of individual bile acids in the serum (**B**). All data are expressed as mean ± SEM. Values with the same letter superscripts mean no significant difference (*p* > 0.05), while values with different letter superscripts mean significant difference (*p* < 0.05). CON, 23 °C with ad libitum intake; HS, 33 °C with ad libitum intake; PF, 23 °C with the same amount to the feed consumed by the HS; BAs, bile acids; TBA, total BAs; PBA, primary BAs, SBA; secondary BAs; GCBA, glycine-conjugated BAs; TCBA, taurine-conjugated BAs; CDCA, chenodeoxycholic acid; HCA, hyocholic acid; UDCA, ursodeoxycholic acid; GCDCA, glycochenodeoxycholic acid; GUDCA, glycoursodeoxycholic acid; TLCA, taurolithocholic acid; TCDCA, taurochenodeoxycholic acid; TUDCA, tauroursodeoxycholic acid; THDCA, taurohyodeoxycholic acid; THCA, taurohyocholic acid.

**Figure 5 animals-10-00359-f005:**
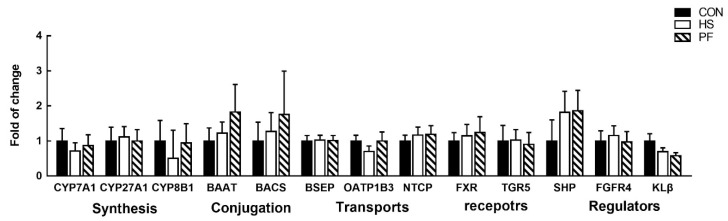
Expression of genes involved in bile acid synthesis, conjugation, transport and regulation in the liver. All data are expressed as mean ± SEM. CON, 23 °C with ad libitum intake; HS, 33 °C with ad libitum intake; PF, 23 °C with the same amount to the feed consumed by the HS; CYP7A1, cholesterol 7α-hydroxylase 1; CYP27A1, sterol 27-hydroxylase; CYP8B1, sterol 12α-hydroxylase; BAAT, bile acid-CoA:amino acid N acyltransferase; BACS, bile acid-CoA synthase, BSEP, bile salt export pump; NTCP, sodium-taurocholate cotransporting polypeptide; OATP1B3, organic anion transporting peptides 3; FXR, farnesoid X receptor; TGR5, membrane-bound G-protein coupled receptor; SHP, small heterodimer partner; FGFR4, fibroblast growth factor receptor 4; KLβ, Klotho beta.

**Table 1 animals-10-00359-t001:** The composition and nutrient levels of diet for growing pigs (air-dry basis) [24,25].

Items	Value, %
Corn	62.55
Soybean meal	27.00
Wheat bran	5.00
Soybean oil	1.00
Limestone	1.50
CaHPO_4_	1.50
NaCl	0.30
L-Lysine-HCl	0.15
Vitamin and mineral premix ^(1)^	1.00
Nutrients ^(2)^	
DE (cal/kg)	3230
CP	18.00
Lys	1.03
Ca	0.95
P	0.74

^(1)^ Premix provided the following per kg of diets: vitamin A 8250 IU, vitamin D3 825 IU, vitamin E 80 IU, vitamin K 4.25 mg, vitamin B1 1.02 mg, vitamin B2 5.20 mg, vitamin B6 2.04 mg, vitamin B12 2.5 mg, biotin 0.2 mg, pantothenic acid 15.3 mg, nicotinic acid 35.7 mg, folic acid 2 mg, Fe (FeSO_4_) 266 mg, Cu (CuSO_4_) 200 mg, Zn (ZnSO_4_) 285 mg, Mn (MnSO_4_)78 mg, I (KI) 0.8 mg, Se (Na_2_SeO_3_) 0.3 mg, choline chloride 600 mg. ^(2)^ Estimated using the NRC (2012) individual dietary ingredients. DE, digestible energy; CP, crude protein; Lys, Lysine; Ca, calcium; P, phosphorus.

**Table 2 animals-10-00359-t002:** Effects of heat stress on serum lipid profiles of pigs.

Items	CON	HS	PF	SEM	*p*-Value
TC, mmol/L	1.97 ^b^	2.26 ^a^	1.77 ^b^	0.19	0.009
HDL-C, mmol/L	0.57	0.52	0.50	0.10	0.302
LDL-C, mmol/L	1.12 ^b^	1.43 ^a^	0.99 ^b^	0.17	0.002
TG, mmol/L	0.41 ^b^	0.66 ^a^	0.34 ^b^	0.14	0.002

TC, total cholesterol; HDL-C, high-density lipoprotein associated cholesterol; LDL-C, low-density lipoprotein associated cholesterol; TG, triglyceride; CON, control group; HS, heat stressed group; PF, pair-fed group. Different superscript letters indicate significant differences among treatment groups in a row (*p* < 0.05).

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
