# Peer review of "Running Head: Heat Affects Cholesterol and Bile Acid Alterations in Cholesterol and Bile Acids Metabolism in Large White Pigs during Short-Term Heat Exposure"

_animals, 2020, doi:10.3390/ani10020359_

Round 1

Reviewer 1 Report

In this study, Fang et al. investigated the alterations in cholesterol and bile acids metabolism in Large White pigs in response to short-term heat exposure. The results showed that short-term heat exposure reduced cholesterol synthesis and promoted cholesterol release to serum and conversion to TCBAs in the liver, which may serve as a mechanism to prevent cellular damage during short term heat stress. This study is well designed and the results are cleared demonstrated. Some minor questions should be amended to improve the manuscript.

1, the figures in the manuscript are not clear, please provide the figures with high resolution.

2, in 2.1, was the chamber temperature of HS group maintained at 33℃ for 24 h every day? Please make clear this point.

3, in 3.4 or Fig. 3, the results showed that the serum secondary BA (SBA) was significantly elevated by HS. So is there any relationship between alteration of SBA and changes of gut microbiota composition? Please provided the related data or discuss more about this issue.

4,what is the possible reasons for the increase in serum TG and decrease in haptic TG. Please discuss more about this.

Author Response

Thanks for all your comments. Please see the attachment.

Reviewer 2 Report

This Article focuses on the effects of short-term heat stress on lipid and cholesterol - bile acid metabolism in large white pigs. This study is very interesting, but there are still some problems in this manuscript and several experiment should be done. I suggest the authors should have a major revision

Abstract

“Compared with CON pigs…respiration rates over 3-fold”, authors should write clearly whether these data are significantly different among groups. “liver taurine-conjugated BAs(TCBA)” should be “TCBAs”,

Results:

The authors should provide high quality figures instead this version. The figure legends should be revised. Authors should give a title to the figure legends, and put the abbreviations to the end of the figure legend. In all the figures, the column are showed with standard deviation or standard error? Please add the information in all the figure legends. Figure 1 legend: “bpm” equals to “breaths”? In addition, please do not follow a sentence after “=”, use “:”. In figure 2 and 4, the black column represents the “CON group” while white column represents “HS group”, but in figure 3, it seems opposite, please revise them. Result 3.4 Authors find HS increased the secondary bile acids (figure 3A), in figure 3B, only showed the concentration of UDCA, so did the authors test other bile acids, such as DCA or LCA? And can authors provide a list how many kinds of bile acids did they test? And how authors define the “primary bile acids”, “secondary bile acids”, “conjugated bile acid” and “unconjugated bile acid’? Figure 4. In figure legends, authors showed several bile acid abbreviations such as “LCA”, “DCA” etc. but I don’t see them in the figure 4 A and B. These abbreviations need to be removed. In Result 3.1, Authors only showed the feed intake, respiratory rate to represent the heat stress, and in Result 3.3, 3.4, 3.5, authors mainly focused on the hepatic health under heat stress, so authors need to test indexes such as AST, ALT, C reaction protein (CRP) ,ROS, caspase 3 or caspase 9 and H&E staining of liver, to elucidate the hepatic health conditions in different treatment groups.

Discussion:

Authors found that TC, TG, LDL-C were increased in HS group, please discuss the relationship between this phenomenon and bile acid changes. In 5th paragraph, authors discussed function of genes such as CYP1A1, CYP21A1, CYP8B1, BAAT, and BACS, which showed no difference in this manuscript, authors should discuss how did this happen, and which other pathways can increase TCBA.

3.Irrelevant part with results should not be over-discuss. For example, altering “In addition to its cooling function, taurine seemed to have a unique mitochondrial protective effect through taurine mediated prevention of damage. Taurine can alleviate the body damages induced by HS, including diminishes oxidative stress and apoptosis” to “What’s more, taurine can alleviate the body damage caused by HS” may be sufficient.

Author Response

(The authors gave the same response as above.)

Reviewer 3 Report

Dear corresponding Authors,

In general, the manuscript is well written and clearly presented. There are only few orthographical errors that need a correction. I value the work that has been done to document the cholesterol and bile acid metabolism in response to heat stress. It is also appreciated that the researchers did the effort to investigate the interference of a reduced feed intake. These results are very valuable to publish.

 I only have minor considerations for the authors, mainly related to the sometimes brief explanation on materials and methods. As there are no line numbers included in the manuscript, my comments and suggestions will be indicated by paragraph (P) number.

Kind regards and good luck with this manuscript.

Author Response

(The authors gave the same response as above.)
